# Effects of Automotive Test Parameters on Dry Friction Fiber-Reinforced Clutch Facing Surface Microgeometry and Wear—Part 3 Tribological Parameter Correlations and Simulation of Thermo-Mechanical Tribological Contact Behavior

**DOI:** 10.3390/polym15051255

**Published:** 2023-03-01

**Authors:** Gábor Kalácska, Roland Biczó

**Affiliations:** Institute of Technology, Szent István Campus, MATE, Páter Károly u. 1., H-2100 Gödöllő, Hungary

**Keywords:** composite clutch friction material, tribo-thermomechanical, modeling, wear correlation, simulation

## Abstract

Correlations among previously determined tribological properties, such as the coefficient of friction values, wear and surface roughness differences of hybrid composite dry friction clutch facings are revealed after pin-on-disk test apparatus examinations under three *pv* loads, where samples are cut from a reference, unused, and several differently aged and dimensioned, used—according to two different trends: dry friction fiber-reinforced hybrid composite clutch facings. In ‘normal use’ facings, increasing specific wear trend is detected as a function of activation energy according to a second-degree function, while a logarithmic trend line can be fitted to the values of the clutch killer facings, showing that even at low activation energy levels, significant (~3%) wear occurs. The specific wear rate also varies as a function of the radius of the friction facing, with the relative wear values measured on the working friction diameter being higher regardless of the usage trend. In terms of surface roughness variation measured in the radial direction, normal use facings show a varying roughness difference according to a third-degree function, while clutch killer facings follow a second-degree or logarithmic trend depending on the diameter (di or dw). From the statistical analysis of the steady-state, we find three different clutch engagement phase characterizing *pv* level pin-on-disk tribological test results for the specific wear of the clutch killer and normal use facings, and significantly different trend curves with three different sets of functions were obtained, showing that the wear intensity can be described as a function of the *pv* value and the friction diameter. In terms of radial direction surface roughness difference, the values of clutch killer and normal use samples can be described by three different sets of function showing the effects of the friction radius and *pv*.

## 1. Introduction

This current work is the continuation of our previous papers [1] of which two articles of three have the same title, referred to as Part 1 [2] and Part 2 [3]. In conventional automotive applications, clutches are used to transmit torque from a power source and a rotating crankshaft to a transmission system and vehicle wheels [4]. All of our work concentrates on fiber-reinforced hybrid composites that are—being the result of a century-long material development process [1]—the most often used materials as friction surfaces of dry friction clutch disks that are positioned between a flywheel and a pressure plate of the clutch in the most common torque transmission applications. As one of the mating surfaces of a friction system, the ever-growing requirements of facings not only cover torque transfer capacity and stable properties under extreme mechanical and thermal load, but are also challenged by lifetime aspects.

Therefore, in the first article, we introduced a novel material identification method [1] providing the mechanical and thermal properties of a certain dry clutch hybrid composite friction material allowing the creation of a complex mechanical material model supplemented by thermal properties.

Since facings should maintain a prescribed minimum coefficient of friction for a certain period of use, commonly given in mileage, in our first follow-up study [2], referred to as Part 1 of this article, we determined surface roughness and wear conditions of the examined hybrid composite friction material during its lifetime following certain automotive tests—according to industrial best practice—described by surface activation energy [2] values. Based on the fact that—assuming constant friction—the *pv* (pressure multiplied by velocity) value is proportionate to the dissipated thermal energy, instead of mileage, surface activation energy was chosen to create a test intensity scale.

Furthermore, our second follow-up and most recent study [3] introduced the tribological behavior of the prescribed material following the best practice field tests through a coefficient of friction values, wear, and surface roughness differences under three different pin-on-disk *pv* loads corresponding to real life application conditions. Therefore, since results were evaluated from pins cut from several differently dimensioned and aged, used, fiber-reinforced hybrid composite dry clutch facings, tests were still characterized by surface activation energy and the previous separation of them into ‘clutch killer’, and ‘moderate’ groups was proven, highlighting the effect of the clutch-operating driver, facing dimensions and test intensity among other parameters.

Our current paper examines possible relations among the previously determined values describing friction composite tribological behavior. However, to determine the correlations among tribological parameters requires further investigations.

Suh and Sridharan gave a mathematical analysis for the relationship between the coefficient of friction and the wear factor of metals for the case of sliding wear. They found agreement between experimentally observed and theoretically predicted values [5]. In terms of wear, in relation to friction, Kato provided a review with the goal of wear control, stating that these tribological parameters are always related to each other when the necessary functions of the tribo-system are well considered [6]. Furthermore, the effect of the surface roughness average of hypereutectic aluminum silicon alloys (with 16 wt. % Si) on the friction and wear was investigated by Al-Samarai et al. They found a correlation between the friction coefficient and hardness, and concluded with the average of the decreasing wear rate and increasing transition stress from high to low wear with increasing surface roughness [7]. On the other hand, a multi-physical signal correlation analysis method evaluating acoustic emission, contact resistance, and frictional force behaviors during dry sliding was proposed to identify the different tribological properties of carbon steel, YG12 carbide, 2A12 aluminum alloy, and H62 brass by Tian et al. [8]. They provided a new method of quantitative identification of the tribological states and the acoustic emission sources during frictional interaction. Meanwhile, the coefficient of friction and wear rate effects were examined by Zawawi et al. to measure the friction and anti-wear abilities of Al2O3-SiO2 composite nanolubricants. They provided a certain volume concentration as the most enhanced composite nanolubricants with an effective coefficient of friction and wear rate reduction compared to other volume concentrations [9].

Friction experiments were carried out by Shi et al. to explore the relationship between the surface topography parameters and friction properties of rough contact interface under fluid dynamic pressure lubrication conditions. They provided a theoretical basis for the functional characterization of surface topography [10]. Surface roughness is a key parameter in tribological systems. Kubiak et al. found that the initial surface roughness significantly affects the coefficient of friction at the transition between partial slip and full slip [11]. Furthermore, Fernandes et al. investigated the effect of cast iron disks with modified surface roughness on the wear rate and coefficient of friction (COF) of a clutch system. They found that the smooth surface cast iron disk reduced the wear rate and contributed to a higher and more stable coefficient of friction level for the engagement stage by not damaging the formed tribofilms and keeping them more stable due to the reduction in contact surface stresses. Modification of the *pv* values toward intensive service conditions influenced the inlet COF variation [12], while Ta et al. established empirical equations that described friction-vibration interactions under dynamic (sliding) lubricated contacts. They found that the mean wear scar diameters exhibited positive correlations with the amplitudes of vibration accelerations [13].

On the other hand, the mechanical, thermal and tribological modeling of the expected behavior of the material can be an essential tool for development as well, providing results more quickly and in a more cost-effective way than testing (considering typically destructive tests), even under different loads. Tribological studies often rely on models in addition to empirical data. Commonly used friction models can be divided into two groups: empirical models and models based on physical equations. The former work better for certain applications, favoring the use of velocity dependence; the latter are able to describe friction before sliding [14].

Among others, Duque et al. evaluated the dynamic Stribeck friction model alongside the Coulomb model to compare their applicability in energy-based simulation of dry clutch coupling during vehicle start-up. Their conclusion was that the Coulomb model, which generates molecular effects and is able to account for surface adhesion, showed more acceptable results [15]. On the other hand, Chu et al. applied the recursive least squares method with the forgetting factor approach (AFFRLS) to determine the friction coefficient of a dry coupling [16].

Grzelczyk and Awrejcewicz performed an experimentally supported mathematical and numerical analysis to investigate the dry clutch contact. In their studies, they used differential and integral wear models to consider the wear properties and elasticity of friction materials. The inclusion of the elasticity of the friction materials allowed a more thorough description of the wear processes and the calculation of the wear distributions over the entire contact surface together with the transmitted torque [17]. In contrast, Li et al. presented a method for predicting the wear of a paper-based wet clutch friction facing subjected to repeated start-up cycles, keeping in mind the fact that the wear behavior of the friction material is closely related to both the thermal degradation of the organic fibers and the stress state at the sliding surface. Predictions from their model were compared with the two-step wear coefficient phenomenon and it was found that the model is able to account for this trend [18].

Furthermore, Abdullah et al. investigated the effect of surface roughness on thermo-elastic behavior through the resulting heat, and developed a new axisymmetric finite element model to study the slip period of the coupling, in which they considered actual surface roughness instead of the general planar surface models. It was concluded that the magnitude and distribution of contact pressure are strongly influenced by the roughness of the contacting pairs [19].

On the other hand, the aim of Fidlin et al. was to create a physical model that can describe and investigate the effect of gaseous particle precipitation on the fading of dry friction couplings. The starting point is the formation of a gas cushion and thus a pressure field, with no influence on the transmittable torque and fading, other than the pressure field. The effects of wear and wear transport were neglected. Three levels of modeling were used: simple rotational symmetry (pressure field inclusion), consideration of axial misalignment (whether it affects the pressure field), consideration of thermoelastic plate deformation (the “recovery effect”). The results showed that the model was able to simulate the phenomenon, with the permeability of the surface appearing to be the determinant of fading: the higher the permeability, the less decomposition occurs. Their assumptions about phase transformation were not confirmed under real production conditions [20].

Though many studies investigated the effect of tribological parameters on each other and the tribological performance as well, and many other papers cover simulation topics focusing on tribological variables, none dealt with effects of applied energy—or surface activation energy—during lifetime on surface morphology, wear and coefficient of friction and their correlations. Furthermore, activation energy dependent coefficient of friction has neither been used in finite element simulation models of dry friction automotive clutches.

Therefore, our aim is to further predict the tribological, lifetime-affected aspects of dry friction hybrid composite clutch facings, focusing on correlations between friction coefficient, specific wear [3], and surface roughness values. The results are aimed at dry clutch finite element contact model development and refinement.

## 2. Materials and Methods

### 2.1. Investigated Materials and the Testing Procedure

For the complex thermomechanical characterization and modeling method with tribological aspects included in our dry friction fiber-reinforced hybrid composite material, Figure 1 provides an illustration highlighting the focus of our current and recent studies. Then, all relevant results are used to create a thermomechanical coupled contact model capable of simulating complex load cases.

The complexity of such materials can be found in the fact that dozens of different components sorted into groups—distinguished by their mechanical role and function—determine their behavior, but even the fiber orientation of glass fiber composites can have an influence on the friction and dry wear behavior [21], not to mention other reinforcing fibers [22] and binders [23] that are effective friction-modifiers. Furthermore, when a component is an industrial secret, throughout identification research is required, as in our case.

Figure 2 summarizes the detailed tribological investigation process detailing the complex structure of the material as well. Facings after production are aged in different field tests, then pins are cut from them via abrasive water jet cutting [24] for pin-on-disk investigations. Highlighted parameters from different stages are evaluated and possible correlations are searched for.

In tribological systems, surface activation energy can transform the surface microgeometry, creating wear, temperature rises, and entropy changes. These are associated with interface material transformation [25]. Induced friction and radial motions during rotation can also lead to unwanted noises and vibrations [26].

Therefore, our parameters describing the surface morphology were surface roughness and wear values after automotive tests run with vehicles equipped with clutch facings made from this examined friction material. Tests varied regarding mileage, operator, field, and many other parameters, such as facing dimensions (Ø228/Ø160/3.5 mm; Ø240/Ø160/3.8 mm; Ø240/Ø155/3.8 mm outer/inner diameters and thickness, respectively), etc., hence the use of surface activation energy to create a common scale for our result visualization. Necessary calculations of the surface activation energy values are also detailed as well in Part 1 of this article along with field test parameters. The results were that wear, and surface roughness vary according to surface activation energy rather than mileage during lifetime, and that facing geometry properties such as thickness, and outer and inner diameter are potential factors governing dry clutch tribological performance [2].

Throughout tribological characterization, to evaluate effects of certain *pv* values, the pin-on-disk test method (carried out according to standard pin abrasion testing—without lubrication, among dry abrasive circumstances—(ASTM G-132) on Ø7 mm hybrid composite test samples) provided the coefficient of friction values and a more reliable comparison of different sample wear and surface roughness difference results through value normalization to unit of surface activation energy. From the measured results, the dynamic friction coefficients were determined as the sum of squares of the surface force components divided by the normal force. From the given parameters detailed in Part 2 of this article, three *pv* levels were chosen, namely minimum, ~0.4 MPa∙m/s, medium, ~0.9 MPa∙m/s and maximum, ~1.8 MPa∙m/s. During clutch actuation or the torque build-up phase—until reaching a certain synchronized rotational velocity to fully transmit the load from the engine toward the transmission—these three values represent the calculated *pv* number, describing the load of the clutch facing its working diameter being the result of a decreasing velocity difference (dv) between the contact pairs (engine shaft velocity of the flywheel versus increasing velocity of the clutch facing) multiplied by the increasing pressure load (p) on the facing as it travels along its cushion deflection displacement. Details of this connection and of the calculation can be found in our previous article [3].

### 2.2. Evaluation Methods of Trioblogical Parameter Correlations

The measurement results of our tribological investigations are average values of several replicates. Each time we determined the standard deviation from the standard deviation square.

The dependence of the studied tribological characteristics on activation energy was investigated by trend line fitting. The strength of the relationship that can be described by a given approximation function is characterized by the so-called coefficient of determination (R^2^). Using the method of least squares we obtained the most common way to fit a model to the measured data. The R^2^ value shows how much of the variance of the dependent variable can be explained by the independent variables [27].

Tribological results often create higher deviation values. On the other hand, results are heavily influenced by test environment parameters, the effect of which cannot be ruled out from the results. Therefore, instead of a best practice value of 95%, we accepted mathematical correlations with about 0.8 R^2^ value as well as good fitting models. In other cases, we plotted the possible trendlines only.

### 2.3. Theory of Model Development

For finite element modeling, the Ansys Workbench version 2021 R2 was used. In the environment provided, coupled analyses are possible. Basically, a standalone static structural or thermal analysis is only capable of the evaluation of the effects of a certain load type at a certain moment in time. Without coupling them, modeling transient time-dependent processes would require constant switching between the two systems, finding ways to carry over displacement field results from one to the other.

Coupled analysis means that the effects of different types of loads (e.g., thermal and mechanical) can be investigated within a single model. The coupled analysis calculates heat distribution, contact pressure, and thermal load induced pressure and force, so that the friction coefficient can be calculated.

When simulating the coupling process of a clutch, it must be taken into account that the friction surface is subjected to a continuously varying load. However, the time dependence of the coupled simulation can be modeled by simulating successive moments of load per second. An example of this is shown in Figure 3 with a setup in Workbench of a 9 s clutch coupling simulation.

### 2.4. Coupled Field Transient Thermomechanical Model Build-Up

The thermomechanical material model of the friction facing was used in a model simulating the installation environment of the friction facing: the coupling between the flywheel and the pressure plate was simulated in a 2D axisymmetric model with the same load as in the automotive tests, recreating the vehicle launch conditions detailed in our previous study. The geometry model is illustrated in Figure 4.

For comparability with the results of the pin-on-disk tests, an evaluation is applied at the time instant of the *pv* values matching reality, at the beginning of the first load second, i.e., 0.3 s, as shown in Figure 5. From the figure, it is also clear that if we neglect the initial relative tapering of the surfaces, the *pv* level effectively becomes diameter-independent.

As for the contacts, between facings, spring elements have been defined to simulate cushion deflection via a displacement dependent force curve. For compression, force load is applied on the pressure plate backward in axial direction. Instead of frictional contact between flywheel and facing, and pressure plate and facing, frictionless behavior has been defined. Friction was considered instead via friction coefficient dependent heat flow calculated via Equation (1) for all facing friction surface parts—energy—as Figure 6 illustrates.
(1)W=M·ω=μ·F·r·ω=μ·A·pv
where:W is the heat flow [W];μ is the coefficient of friction [-];A is the corresponding surface [m^2^];*pv* is the surface pressure multiplied by velocity [MPa m/s]

Each step, the rotational velocity difference decreases, and increasing energy is applied, until the sum of the surface activation energy reaches the level of the simulated test scenario energy level. Following each step of the simulation, contact pressure values are evaluated to calculate new *pv* levels that determine the coefficient of friction for the next step chosen from our measurement data published in our previous study.

Therefore, the model is capable of simulating—or substituting—real life field tests that are used to obtain a clear picture in a relatively quick way (days, weeks, and year based on test type [3]) of tribological performance changes during lifetime. Chapter 4 provides an example of its use.

## 3. Correlations among Dry Friction Hybrid Composite Tribological Parameters

In our previous study, we introduced pin-on-disk (PoD) investigations that were used for creating different wear stages of the examined fiber-reinforced hybrid composite friction material. Those wear stages recreate conditions among which different surface activation energy values were applied that the facings had endured in automotive tests. We compared relative result values to highlight dependence on the different parameters. Therefore, results such as wear and surface roughness difference (compared to results after preliminary field tests) were normalized to activation energy.

In this paper, we continue our investigation concentrating on similarities between different result trends searching for correlations. Values we examine can be found in our previous paper, Part 2 of this article [2]. The measured characteristics, such as wear and surface roughness values, are plotted along the surface activation energy scale. Using trend line fitting, we seek correlations between the results measured after the automotive tests and those obtained from the pin-on-disk tests for the results of the two trends established from the measurements separately.

In tribological practice, low R^2^ values are relatively common. In such cases, we only showcase possible trendlines. Equations are detailed only for those models, where R^2^ values reach ~0.8.

### 3.1. Analysis of Wear Values—Thickness Reduction Versus the Original Thickness

The thickness values of the tested facings were 3.5 mm or 3.8 mm. The wear values measured after the automotive tests are compared to the thickness of the facings in 100% form. The results are shown in Figure 7 and Figure 8.

It can be seen that the normal use facings show increasing wear according to a quadratic power function, while the values for the clutch killer-group facings show a logarithmic trend line: even at low activation energy levels, significant wear occurs. In both cases, the relative wear values measured on the friction diameter are higher, but the difference between the wear values of the diameters is smaller for normal usage, which can be explained by the smaller tapering of the friction surfaces during lifetime.

After pin-on-disk tests, the specific wear values normalized to the surface activation energy unit are shown in Figure 9 or Figure 10, depending on the use trend.

The logarithmic relationship can be observed, but with a negative factor at maximum *pv* load, regardless of the cutting diameter, i.e., as the surface energy increases, the wear decreases under maximum *pv*. This is presumably due to the third body phenomena, which reduces the intensity of wear formation on the friction surface at certain high loads.

By normal usage for both cut-out diameters (di: internal, dw: friction), the trend for almost all *pv* load levels is characterized by unrelenting clutch wear below maximum *pv*. Conversely, under sufficiently high *pv*, the clutch killer use gives results similar to normal use. This presumably indicates, in line with the previous results, the wear reducing effect of the third body phenomenon.

### 3.2. Analysis of Ra Surface Roughness Values in Radial Direction

The radial direction surface roughness values after the automotive tests are compared to the surface roughness of unused facings in 100% form. The results are shown in Figure 11 and Figure 12.

From the analysis, it can be seen that in terms of surface roughness variation in radial direction, the values of the normal use facings show a third-order trend at both cut-out diameters. In the case of clutch killer usage, the logarithmic relationship observed for the wear trends appears only on the inner diameter, the values of the friction diameter follow the second order trend observed for the wear trends of the normal use facings. Regardless of the use trend, the decrease in Ra values is smaller in absolute values for the friction diameter compared with the inner diameter.

After pin-on-disk tests, results are illustrated in Figure 13 and Figure 14.

Of clutch killer trend facing samples, the trends in roughness variation due to different *pv* levels follow three different types of functions: the results for internal diameter specimens are described by linear and quadratic equations, and the post-pin-on-disk values for specimens cut from friction diameter are described by linear and logarithmic equations. Additionally, for normal use facings, results show a strong *pv*-dependence with linear and logarithmic trends.

## 4. Dry Friction Hybrid Composite Tribological Behavior Investigation Results Use

The simulation model is capable of taking effects of wear into account. One direction for further development could be the parameterized automation of wear by creating simulations with arbitrary wear intensities.

Another application of the model highlights the oversized state of currently used conventional vehicle clutches, i.e., their cost reduction potential. The currently used sizing procedures consider the friction coefficient as a constant value up to a certain temperature level, without taking into account the *pv*-dependence.

Figure 15 illustrates that heat loads calculated with real, *pv*-dependent COF values result in lower temperatures, so that the currently used calculation methods oversize the pressure plate, for example.

## 5. Conclusions

Tribological properties, such as wear and surface roughness difference values of dry friction fiber-reinforced hybrid composite clutch facings, were examined after automotive best practice and standard pin-on-disk tests, as a follow-up of our recent studies. Results were depicted along the surface activation energy scale instead of mileage. Correlations were searched for among the tribological properties and other influencing factors.

From the statistical analysis of the wear test results evaluated after the automotive tests with all the clutch shifts, considered normal use facings showed an increasing specific wear trend as a function of activation energy according to a second-degree function, while a logarithmic trend line was able to be fitted to the values of the clutch killer facings, showing that even at low activation energy levels significant wear occurs. The radius of the friction facing turned out to have an influence on the specific wear rate. The diameter where friction occurs also had an influence: the relative wear values measured on the friction diameter were always higher regardless of the usage trend.

In terms of surface roughness variation measured in the radial direction after the automotive tests, normal use facings showed a varying roughness difference according to a third-degree function, while clutch killer facings followed a second-degree or logarithmic trend depending on the diameter (di or dw).

From the statistical analysis of the steady-state three different clutch engagement phase characterizing *pv* level pin-on-disk tribological test results of samples cut from facings worn in automotive tests, it can be concluded that for the specific wear of the clutch killer and normal use facings, significantly different trend curves with three different sets of functions were able to be obtained. This indicates that the wear intensity varies during the clutch engagement phase, i.e., it can be described as a function of the *pv* value and also as a function of the sample cutting, i.e., the friction diameter. In terms of radial direction surface roughness difference, the values of the samples of the two different groups of facings in different modes of use can be described by three different sets of functions, which also shows the dependence on the friction radius and *pv* promoting them to be significant parameters.

Finally, a demonstration highlighted the practical use of surface activation energy (lifetime)-dependent tribological parameters. Complex tribo-thermomechanical contact of dry friction woven fiber-reinforced hybrid composite clutch facings modeled by coupled thermomechanical simulation using our previously created orthotropic material model takes the effects of the *pv*- and energy dependent friction coefficient into account. This model can be further developed into an automatic and parameterized wear simulation model, which is the aim of our future studies.

## Figures and Tables

**Figure 1 polymers-15-01255-f001:**
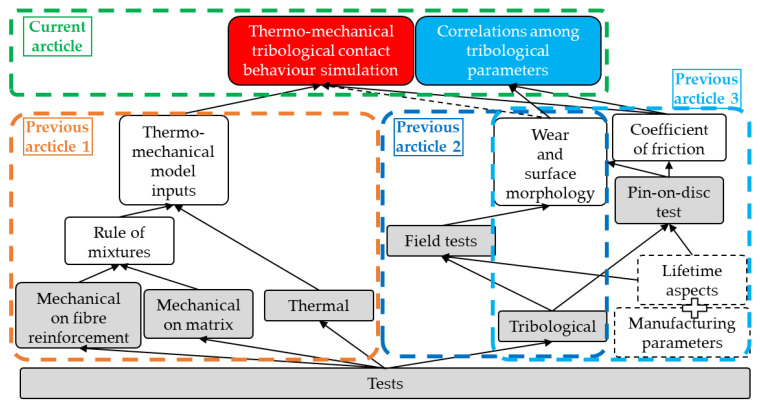
Tribo-thermomechanical characterization and modeling method.

**Figure 2 polymers-15-01255-f002:**
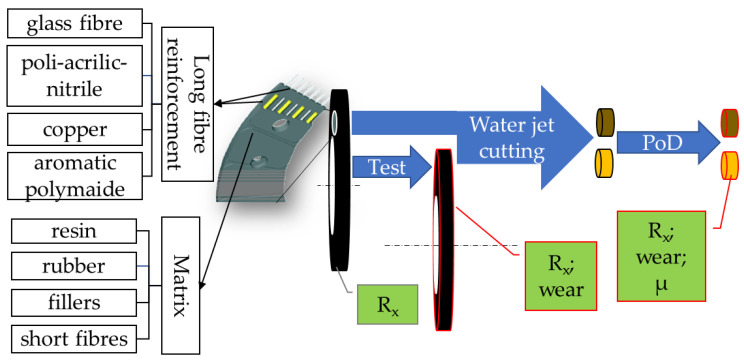
The hybrid composite material components and the tribological investigation process. where R_x_ is R_a_, R_z_ and R_max_.

**Figure 3 polymers-15-01255-f003:**
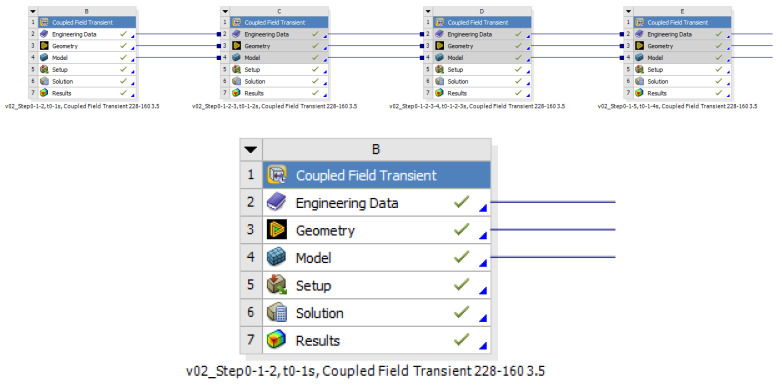
Workbench boxes for each second during the thermomechanical contact build-up.

**Figure 4 polymers-15-01255-f004:**
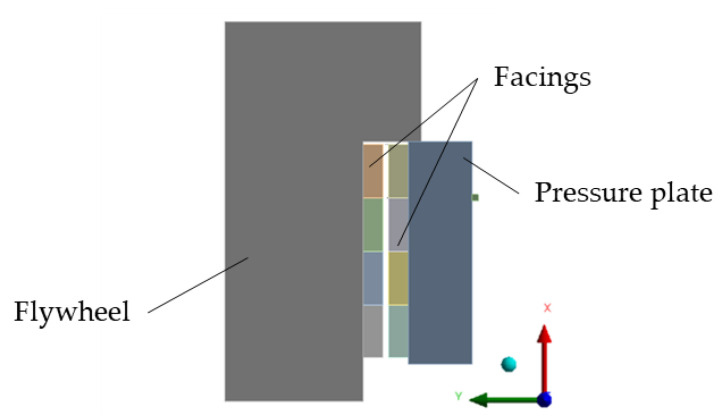
Geometry used in coupled 2D simulation: friction facing pair between simplified flywheel and pressure plate models.

**Figure 5 polymers-15-01255-f005:**
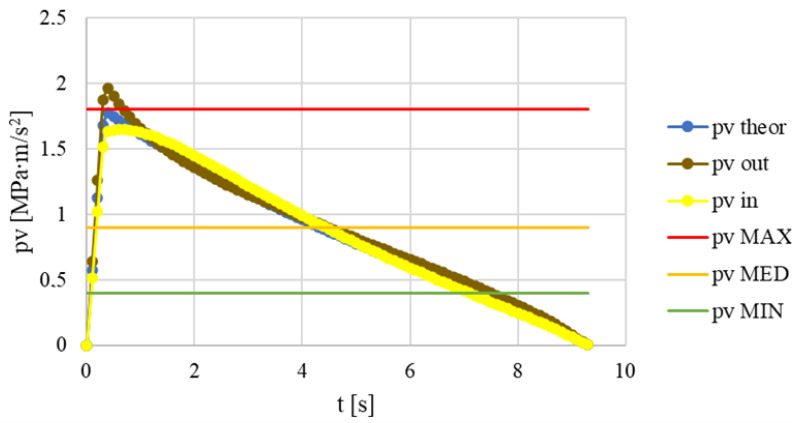
*pv* levels during simulation model operation at inner (*pv* in), outer (*pv* out) and middle (*pv* theory) diameters and MAX-MED-MIN *pv* values used for pin-on-disk measurements.

**Figure 6 polymers-15-01255-f006:**
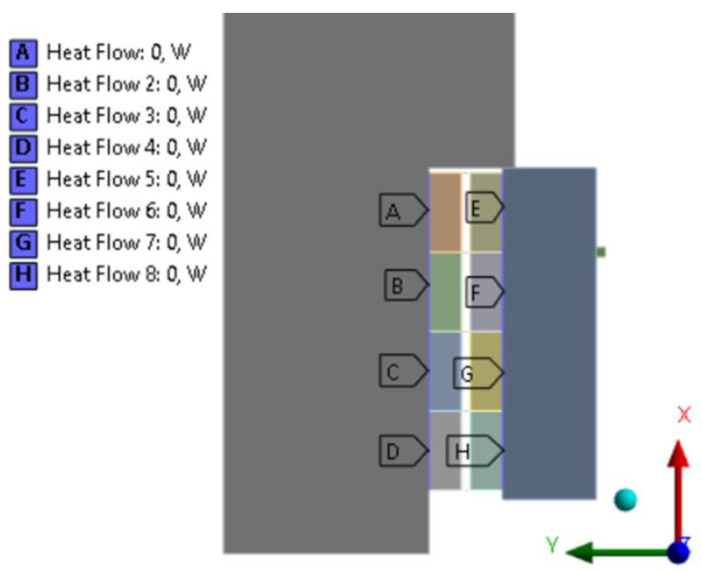
Heat flow applied among mating friction surfaces modeling energy transfer.

**Figure 7 polymers-15-01255-f007:**
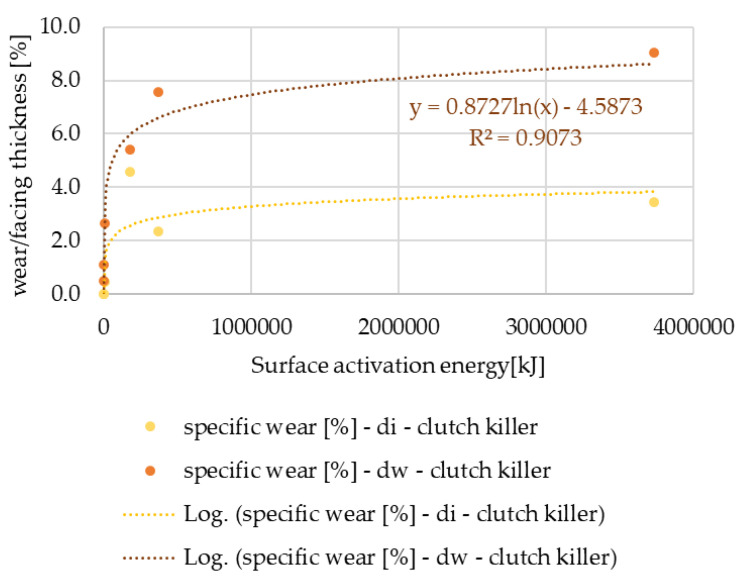
Wear % values versus facing thickness as a function of surface activation energy—clutch killer facings.

**Figure 8 polymers-15-01255-f008:**
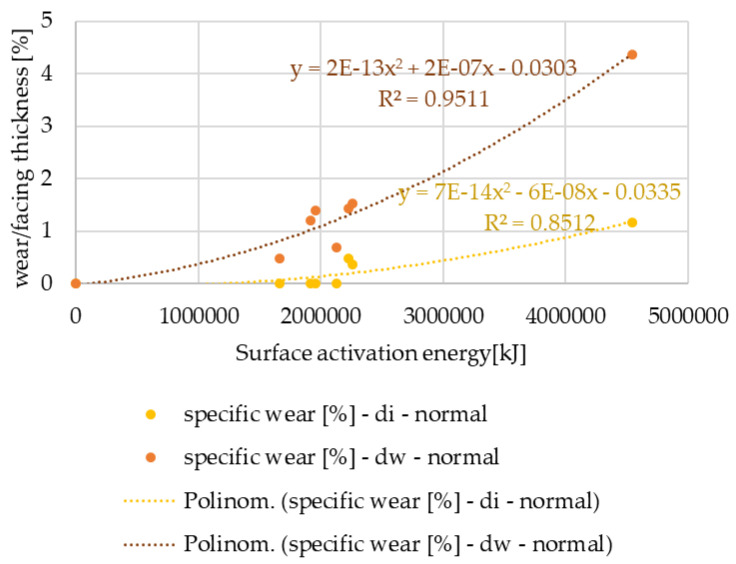
Wear % values versus facing thickness as a function of surface activation energy—normal use facings.

**Figure 9 polymers-15-01255-f009:**
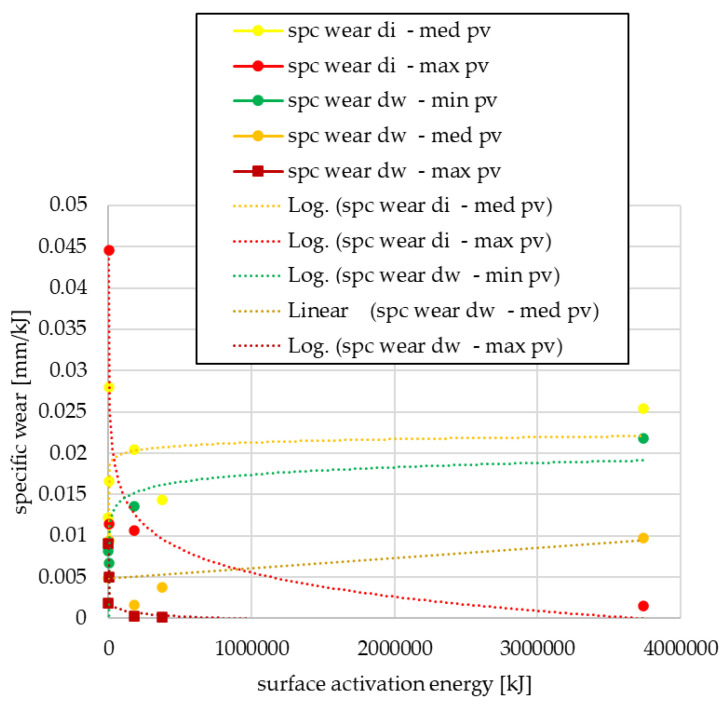
Specific wear values as a function of surface activation energy—clutch killer facings.

**Figure 10 polymers-15-01255-f010:**
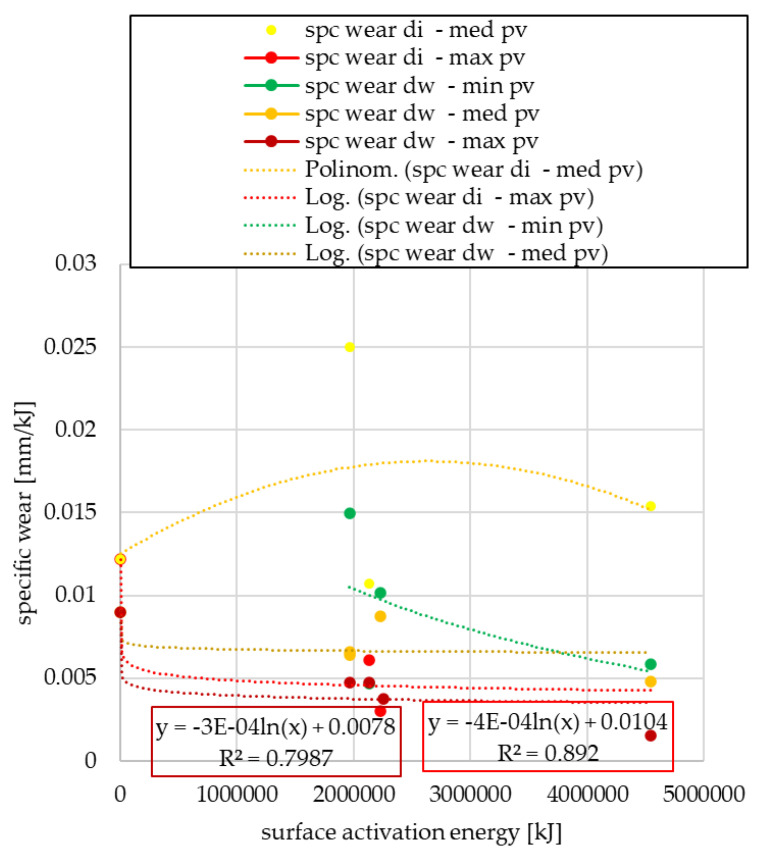
Specific wear values as a function of surface activation energy—normal use facings.

**Figure 11 polymers-15-01255-f011:**
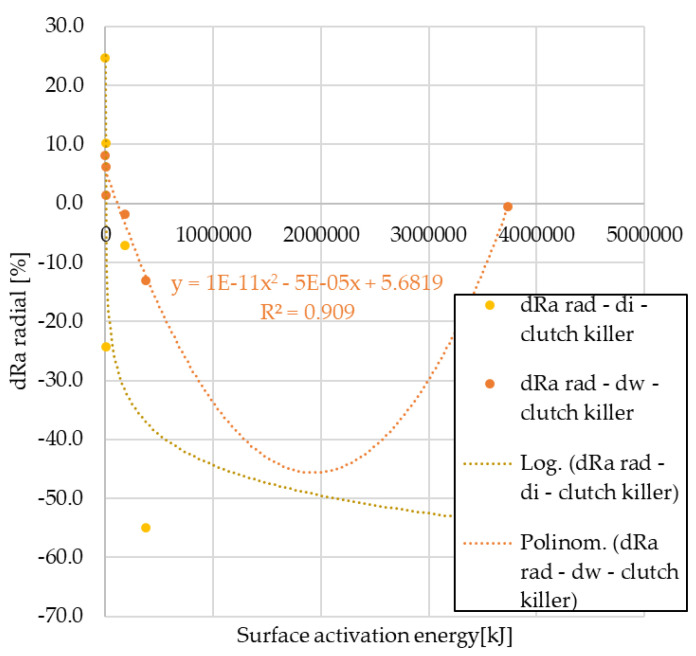
Variation of radial Ra values as a function of activation energy—clutch killer facings.

**Figure 12 polymers-15-01255-f012:**
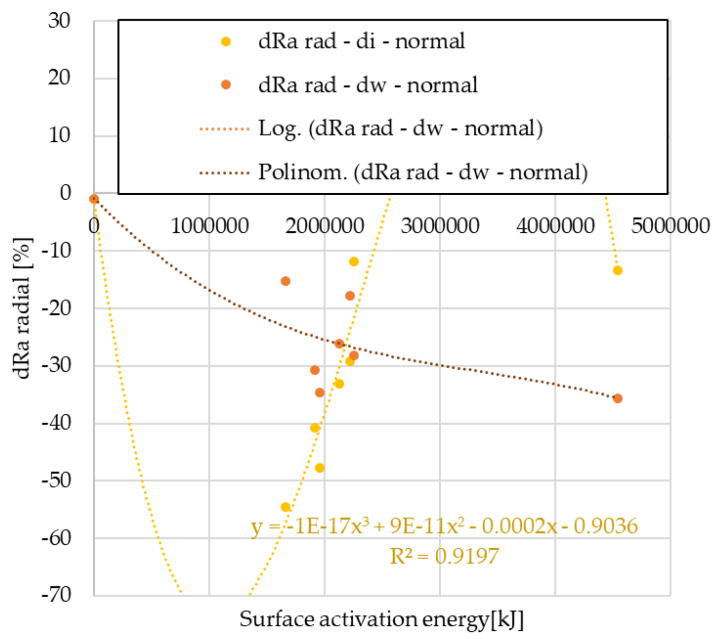
Variation of radial Ra values as a function of activation energy—normal use facings.

**Figure 13 polymers-15-01255-f013:**
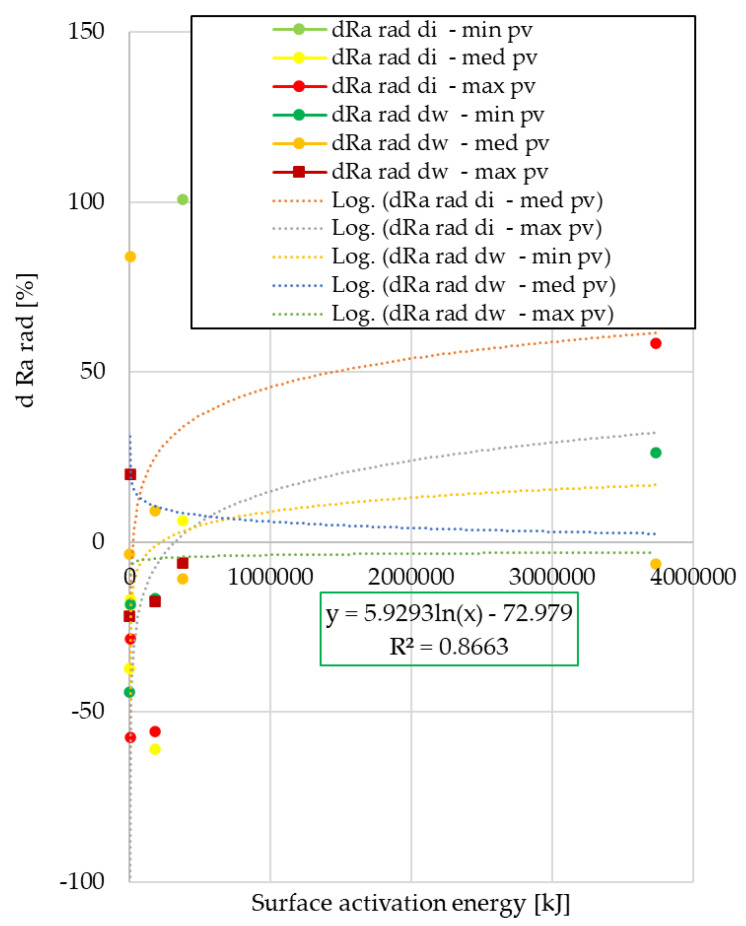
Variation of radial Ra values after pin-on-disk as a function of surface activation energy—clutch killer facings.

**Figure 14 polymers-15-01255-f014:**
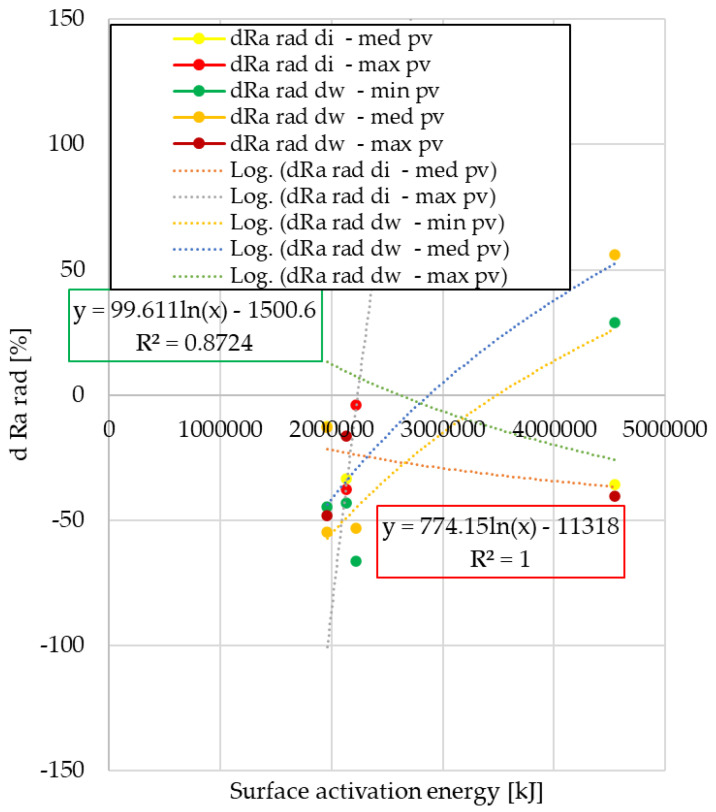
Variation of radial Ra values after pin-on-disk as a function of surface activation energy—normal use facings.

**Figure 15 polymers-15-01255-f015:**
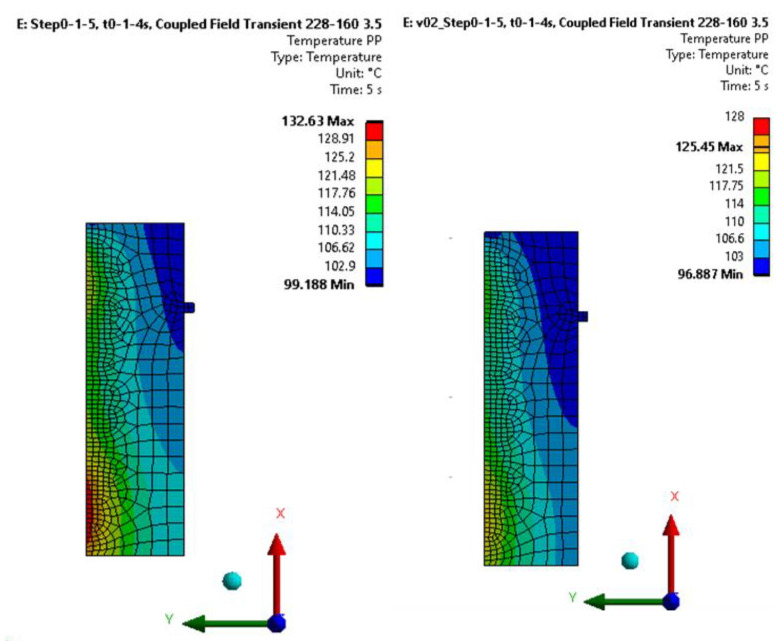
Temperature distribution on the trace plate: (**left**): constant COF, (**right**): pv-dependent COF.

## Data Availability

Not applicable.

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
