# Peer review of "Effects of Automotive Test Parameters on Dry Friction Fiber-Reinforced Clutch Facing Surface Microgeometry and Wear—Part 3 Tribological Parameter Correlations and Simulation of Thermo-Mechanical Tribological Contact Behavior"

_polymers, 2023, doi:10.3390/polym15051255_

Round 1
Reviewer 1 Report
In this manuscript the wear of clutch facings is investigated focusing on the correlation to the surface activation energy using the previously in paper ‘Part 2’ obtained data. Different relations for two types of facings - normal use facings and clutch killer facings are observed. In Figure 7 and Figure 8 one can clearly see these dependences. However, in later figures, Figure 10 – Figure 14, the logarithmic and polynomial functions do not fit the data well sometimes, for example the yellow and green dashed lines in Figure 10. The curves and functions in figures should be explained in captions or context clearly: symbols are experimental results, but why there seems two fittings (dashed lines in Figure 11 and Figure 12)?
Another question is about the friction coefficient: It is stated that ‘Friction was considered instead via friction coefficient dependent heat flow calculated via Equation 1 for all facing friction surface parts’ and ‘frictionless behavior has been defined’, but the friction coefficient is still given in simulation as shown in Equation (1)? I don’t understand this part.
Reviewer 2 Report
The authors studied influences of automotive test parameters on dry friction
fiber-reinforced clutch facing surface microgeometry and wear. The authors' work is valuable. The reviewer thinks that the manuscript can be accepted before the following several issues are improved or clarified.
1. The English language needs to be improved.
2. The definition of the surface activation energy should be presented.
3. The title of the paper contains "automotive test parameters", However, the automotive test parameters were not presented in the paper. Please explain. Or the authors can change the title of the manuscript.
4. What is the definition of "Specific wear"?
